Phytochemical profiling and antioxidant activity assessment of Bellevalia pseudolongipes via liquid chromatography-high-resolution mass spectrometry

http://orcid.org/0000-0001-7529-3395 Yolbaş İdris idrisyolbas@gmail.com
Türk Telekom Science High School , Siirt, Siirt , Turkey
Lenardão Eder
Electronic publication date: 2024 Sep 13
Publication date: 2024
Volume: 12
Electronic Location ID: e18046
Received 2024 May 24; Accepted 2024 Aug 14
Copyright: © 2024 Yolbaş
Copyright year: 2024
Copyright holder: Yolbaş
License: This is an open access article distributed under the terms of the Creative Commons Attribution License, which permits unrestricted use, distribution, reproduction and adaptation in any medium and for any purpose provided that it is properly attributed. For attribution, the original author(s), title, publication source (PeerJ) and either DOI or URL of the article must be cited.
License URL: https://creativecommons.org/licenses/by/4.0/

Keywords: Phytochemicals, Bellevalia pseudolongipes (Asparagaceae), Antioxidants

Funding: The author received no funding for this work.

==============================
Background

Plant-derived drugs are often preferred over synthetic drugs because of their superior safety profiles. Phenolic compounds and flavonoids—major plant components—possess antioxidant properties. Limited research has been conducted on the bioactive compounds and biochemical properties of Bellevalia pseudolongipes (Asparagaceae), an important pharmacological species endemic to Turkey. Therefore, the chemical composition and antioxidant properties of B. pseudolongipes were investigated in this study.

Methods

The chemical composition of B. pseudolongipes was analyzed using liquid chromatography-high-resolution mass spectrometry, and radical scavenging and antioxidant activities were evaluated using DPPH (2,2-diphenyl-1-picrylhydrazyl) and ABTS (2,2′-azinobis(3-ethylbenzothiazoline-6-sulfonic acid)) tests.

Results

Thirty-eight compounds were identified, including trans-cinnamic acid, caffeic acid, vitexin, schaftoside, orientin, and narirutin. B. pseudolongipes showed high antioxidant activity in antioxidant activity tests.

Conclusion

These findings provide novel insights into the potential utility of B. pseudolongipes in the pharmaceutical, food, and cosmetics industries, highlighted by its significant antioxidant capacity.

Introduction

Plants play a fundamental role in sustaining human life, serving as valuable resources for pharmaceuticals, cosmetics, nourishment, and household cleaning products (Njume, Afolayan & Ndip, 2009; Vdzhao et al., 2021; Zhao et al., 2021; Otuokere et al., 2022; Qi et al., 2023; Yolbaş, 2024). Historically, humans have harnessed the medicinal properties of plants, leading to the continuous discovery and use of plants with therapeutic attributes. In the Mesopotamian civilization, approximately 250 herbal remedies were routinely used. This number increased to 600 in ancient Greece and surged to 4,000 during the Arab–Persian epoch. By the early 19th century, approximately 13,000 plant species had been extensively utilized for therapeutic purposes (Farnsworth & Soejarto, 1985), and approximately 25% of the prescription medications in developed nations contain plant-derived compounds (Khan & Ahmad, 2019). Herbal plant extracts known for their antioxidant activity play a crucial role in preventing diseases associated with oxidative stress (Owen et al., 2000). These properties can be attributed to plant secondary metabolites, such as phenolic and flavonoid compounds (Keskin et al., 2018). Antioxidant-rich foods help protect the human body from the detrimental effects of oxidative stress induced by free radicals, playing a critical role in disease prevention (Gülçin, 2020). Such findings indicate that antioxidant activity is closely linked to the phenolic content, implying that plants with high phenolic content are rich in antioxidants and therefore potentially effective at disease prevention.

Phenolics are among the most abundant molecular groups, with at least 10,000 different compounds possessing and exhibiting beneficial bioactivities (Rasouli, Farzaei & Khodarahmi, 2017). Recent studies have highlighted the potential side effects of artificial food additives, increasing interest in using natural preservatives (Carocho et al., 2014). Medicinal plants are extensively used in the treatment of many human and animal diseases nowadays (Sharifi-Rad et al., 2015). These compounds possess antioxidant, antimicrobial, antidiabetic, antimutagenic, anti-inflammatory, and anticarcinogenic properties (Owen et al., 2000). The well-known side effects of synthetic drugs, especially in treating neurodegenerative diseases and cancers, underscore the importance of characterizing active ingredients in medicinal plants and determining their effective bioactivities. Using medicinal plants can significantly reduce side seffects (Verma & Singh, 2008).

The preference for plant-derived drugs over chemically synthesized drugs is largely because of their enhanced safety profiles. The rising incidence of antibiotic resistance and the advent of multidrug-resistant strains present global health challenges (Adoum, 2016). Consequently, there is an imminent demand for plant-derived novel agents. However, these compounds may differ according to plant species, thus warranting distinct investigations for each species.

Bellevalia pseudolongipes, an indigenous plant species of Turkey, is particularly widespread in the regions of Siirt and Mardin. This plant grows in steppe areas at altitudes of 950–1,150 m within the Irano–Turanian floristic region. The genus Bellevalia comprises 23 species, 13 of which are endemic to Turkey (Karabacak, Yildirim & Martiïn, 2014). It is pharmacologically important owing to its high bioactive homoisoflavonoid content (Lin, Liu & Ye, 2014).

Phenolic compounds, characterized by their benzene ring structure, are abundant in plants and vary among species, imparting color to fruits and flowers and conferring protection against environmental stress. Polyphenolic compounds and flavonoids exhibit antioxidant properties, mitigating the adverse effects of free radicals and reducing lipid peroxidation, thereby extending the shelf life of foods (Gülçin, 2010; Al-Jaber, Shakya & Elagbar, 2020; Bobrovskikh et al., 2020; Rahmawati, Ardana & Sastyarina, 2021; Gurung et al., 2022; Thuy et al., 2023; Tura et al., 2023). Liquid chromatography–high-resolution mass spectrometry (LC-HRMS) is a bioanalytical tool used to analyze natural products found in food and herbal sources. This method can overcome various analytical limitations and is used to detect and characterize new secondary metabolites of natural products frequently manufactured in limited quantities (Aydoğan, 2020). LC–HRMS is also suitable for analyzing the non-volatile chemical composition of plant extracts. However, data on the bioactive compounds and biochemical properties of B. pseudolongipes are limited. To date, no study has used LC–HRMS to detect phenolic compounds in B. pseudolongipes.

Therefore, the aim of the present study is to comprehensively analyze the chemical composition, radical scavenging and antioxidant activities of B. pseudolongipes. The findings presented herein could facilitate future studies on the potential applications of B. pseudolongipes in plant pharmacology, food processing, and cosmetics.

Materials and Methods

Collection of plant materials

B. pseudolongipes samples (Fig. 1) were collected in the Siirt province of Turkey (37°57′49″N, 41°50′38″E) toward the end of April 2023. The collected samples were dried in shade at 24 °C. Each plant part—including the bulb, stem, leaf, and flower—was collected whole, dried, and then homogeneously powdered in a grinder for analysis.

Figure 1 Photograph of Bellevalia pseudolongipes featuring the characteristic bell-shaped flowers and distinctive morphology.

Antioxidant activity analysis

Extract preparation

Five milliliters of 75% methanol containing 0.1% phosphoric acid was added to 0.2 g of the powdered plant sample; the resultant was mixed thoroughly and homogenized using an Ultra Turrax homogenizer (MS3-MaxiHomo35; Maxiab Biotechnology, Staufen, Germany) at 600 rpm for 30 s. The homogenate was centrifuged (Archer LC-05A; Los Angeles, CA, USA) at 2,500 rpm (3,600 ×g) for 10 min at 24 °C. Subsequently, the supernatant was incubated for 15 min in an ultrasonic water bath at 25 °C and transferred to a tube for antioxidant activity assessment. This extraction process was repeated twice, and the resulting extracts were combined.

2,2-Diphenyl-1-picrylhydrazyl radical scavenging activity

The antioxidant activity of the extract was evaluated using the 2,2-Diphenyl-1-picrylhydrazyl (DPPH) radical scavenging assay, following the method outlined by Bersuder, Hole & Smith (1998), with minor modifications. Briefly, 0.1 mL of the extract was placed in a 10 mL tube, and methanol was added such that the total volume reached 10 mL. Subsequently, 0.1 mL of the methanol-diluted extract was mixed with 3.9 mL of a 60 µM DPPH-methanol solution and incubated for 30 min at 24 °C. The absorbance of the samples was measured at 517 nm using a Jasco/V-730 spectrophotometer (Istanbul, Turkey). The Trolox solvent served as the blank, whereas the DPPH solution served as the control.

DPPH calibration was performed using solutions of different concentrations (20, 40, 60, 80, and 100 µg/mL). A DPPH calibration curve was constructed, and the half-maximum inhibitory concentration (IC50) was calculated. A Trolox curve was constructed using Trolox standard solutions at different concentrations, and the IC50 was calculated as the Trolox-equivalent from the Trolox and DPPH curves. The inhibition percentage was determined using Eq. (1) as follows:

(1) %inhibition=[(Ak−A0)/Ak]×100

where Ak represents the absorbance of the control sample without antioxidants, and A0 represents the absorbance of the sample containing antioxidants. Using these values, an inhibition curve was plotted, and IC50 was determined as the concentration at which the sample scavenged 50% of the DPPH radicals.

ABTS radical scavenging assay

In our study, we employed the ABTS clearance determination method proposed by Re et al. (1999). Initially, a 7 mM ABTS solution was prepared in a 140 mM potassium phosphate buffer and allowed to stand for 12–16 h at room temperature in the dark to ensure the conversion of ABTS molecules into their radical cation form. Subsequently, this solution was diluted with ethanol to achieve an absorbance of 0.70 ± 0.02 at a wavelength of 734 nm. For sample preparation, 25 μL of the sample solution was mixed with 2 mL of the 7 mM ABTS solution and reacted in the dark for 6 min. The antioxidant capacity of the sample was then quantified based on its interaction with the ABTS radical cations, with results expressed in Trolox equivalents and used for quantitative determination of antioxidant capacity. This method provides an objective evaluation of antioxidant properties using a standardized protocol.

LC–HRMS analysis of the plant extract

Sample preparation

Ten milligrams of powdered plant sample was dissolved in 10 mL of a 1:1 v/v mixture of methanol and water and then passed into a 1.5 mL vial via a 0.22 µm polytetrafluoroethylene syringe filter (25 mm diameter).

Chromatography and high-resolution mass spectrometry conditions

The analysis was conducted using a Phenomenex Gemini 3 µm NX-C18 110 Å (100 mm × 2 mm) column (Phenomenex, Torrance, CA, USA). The column furnace was set to 30 °C. Elution was conducted using 2% (v/v) glacial acetic acid in ultrapure water obtained from a GFL 2004/Human power 1 ultrapure water system (Tokyo, Japan) as mobile phase A and 99.9% pure methanol (Sigma-Aldrich, St. Louis, MO, USA) as mobile phase B. Separation was performed with a flow rate of 0.3 mL/min, a sample injection volume of 20.0 µL, and an analysis duration of 20 min. In the analysis, an Orbitrap HRMS system (Exactive Plus™; Thermo Fisher Scientific, Waltham, MA, USA) equipped with a heated electrospray ionization interface was used and operated in positive (full MS/all-ion fragmentation (AIF)) and negative (full MS/AIF) modes. The specific analysis parameters were as follows: automatic gain control target, 3e6; spray voltage, 3.5 kV; S-lens RF level, 50; maximum IT (limit of time for accumulating ions per scan event), 2 ms; ionization interface sheath gas flow rate, 35 mL/min; auxiliary gas temperature, 350 °C; auxiliary gas flow rate, 7 mL/min; MS scan, 60–800 m/z; capillary temperature, 350 °C; resolution, 17,500×; collision energy/step, <25 V.

LC–HRMS

LC–HRMS analysis was performed using the high-resolution MS composition function, a DIONEX UltiMate 3000RS autosampler, an LC system equipped with a DIONEX UltiMate 3000RS pump, a DIONEX UltiMate 3000RS column furnace, and an Exactive Plus Orbitrap (Thermo Fisher Scientific, Waltham, MA, USA) equipped with a heated electrospray ionization interface. Calibration of the Orbitrap-LC–MS instrument was conducted using negative (Pierce™ Negative Ion Calibration Solution; Rockford, MA, USA) and positive (Pierce, LTQ Velos ESI Positive Ion Calibration Solution) calibration solutions through an automated syringe injector (Thermo Fisher Scientific, Waltham, MA, USA). Simultaneous LC and MS analyses were conducted during LC–HRMS using TraceFinder 3.2 (Thermo Fisher Scientific, Waltham, MA, USA), and data acquisition and processing were performed using the Xcalibur software version 2.1.0.1140 (Thermo Fisher Scientific, Waltham, MA, USA).

Results and discussion

Phytochemical content findings

A total of 88 phytochemical compounds were utilized as standards in the LC–HRMS analysis of B. pseudolongipes extracts, identifying 38 compounds with high-intensity peaks (Table 1). In comparison, Balos (2021) detected 26 compounds in B. pseudolongipes extracts using LC–MS/MS. Moreover, an investigation into the composition of Bellevalia sasonii using LC–MS/MS revealed that of 53 phytochemicals utilized as standards, 27 were absent in all extracts, 26 were present in at least one extract, and five were consistently present in all plant parts, including bulbs, flowers, leaves, and stems (Tekin, 2022). Demirci (2014) reported the detection of flavone, tannin, and cardiotonic glycosides in several Bellevalia species (B. macrobotrys, B. tauri, and B. gracilis). In Bellevalia flexuosa onion extract, El-Elimat et al. (2018) identified four new compounds belonging to the homoisoflavonoid group. Additionally, a study on the endemic Bellevalia mauritanica plant qualitatively determined 91 phytochemicals, with caffeic acid and vanillin being prominent components (Ouelbani et al., 2020). These findings suggest that the diversity and concentrations of phytochemical compounds may vary among different plant samples and that the analytical methods can have a considerable impact on the findings.

Table 1 Phenolic compound profile of Bellevalia pseudolongipes.

No.	Phenolic compound	Content (mg/kg)	No.	Phenolic compound	Content (mg/kg)	
1	Luteolin-7-O-β-D-glucuronide (luteolin-7-O-glucuronide)	N/F	45	Acacetin (5,7-dihydroxy-2-(4-methoxyphenyl)-4H-chromen-4-one)	N/F	
2	2,4-Dihydroxybenzoic acid (beta-reconcile acid)	N/F	46	Hyperoxide (quercetin 3-D-galactoside)	43.656	
3	3,4-Dihydroxybenzaldehyde (protocatechuic aldehyde)	24.666	47	Isoorientin	1,308.652	
4	3,4-Dihydroxyphenylacetic acid (DOPAC; homoprotocatechuic acid)	35.353	48	Isorhamnetin (quercetin 3′-methyl ether)	N/F	
5	3-Hydroxybenzoic acid (3-HBA)	0.186	49	Isoquercitrin (quercetin 3-glucoside)	43.656	
6	3-Hydroxyphenylacetic acid (3-HPA)	221.558	50	Kaempferide	N/F	
7	4-Hydroxybenzoic acid	8.709	51	Kaempferitrin	N/F	
8	4-Hydroxy-3-methoxyphenylacetic acid (homovanillic acid)	27.639	52	Kaempferol	N/F	
9	Afzelin (kaempferol 3-rhamnoside)	3.674	53	Kuromanin (cyanidin 3-glucoside chloride)	N/F	
10	5,7-Dihydroxy-3-(4-hydroxyphenyl)-4H-chromen-4-one (genistein)	N/F	54	Leucoside (kaempferol 3-sambubioside)	N/F	
11	Apigenin 7-glucoside	15.696	55	Liquiritigenin	N/F	
12	Apigenin 7-glucuronide	N/F	56	Kaempferol 3-rutinoside, kaempferol 3-O-β-rutinoside (nicotiflorin)	N/F	
13	Apiin (apigenin-7-(2-O-apiosylglucoside))	79.875	57	Luteolin	124.969	
14	Arbutin	85.031	58	Luteolin 7-rutinoside	32.313	
15	Astragalin (kaempferol 3-glucoside)	N/F	59	Luteoloside (luteolin 7-glucoside)	78.76	
16	Benzoic acid	48.755	60	Narcissin (narcissoside, isorhamnetin 3-rutinoside)	N/F	
17	Caffeic acid	259.684	61	Naringenin	N/F	
18	Caffeic acid phenethyl ester (CAPE)	N/F	62	Naringin	N/F	
19	Catechin (cianidanol)-p	N/F	63	Narirutin (naringenin-7-O-rutinoside)	472.752	
20	Chlorogenic acid	N/F	64	Neohesperidin	0.009	
21	Ethyl 3,4-dihydroxybenzoate (protocatechuic acid ethyl ester)	N/F	65	Liquiritin (4′,7-dihydroxyflavanone 4′-glucoside)	N/F	
22	Coumaric acid (trans-3-hydroxycinnamic acid)	2.503	66	Orientin	1,308.652	
23	Daidzin	N/F	67	Phloridzin	N/F	
24	Diosmetin (luteolin 4′-methyl ether)	12.85	68	Procyanidin B2	7.652	
25	Doxorubicin hydrochloride	N/F	69	3,4-Dihydroxybenzoic acid (Protocatechuic acid)	6.962	
26	Ellagic acid	74.263	70	5,7-Dihydroxy-2-phenyl-4H-chromen-4-one (chrysin)	N/F	
27	Emodin	N/F	71	Quercetin	8.241	
28	Epigallocatechin	N/F	72	Quercetin 3-rutinoside 7-glucoside	N/F	
29	Epigallocatechin gallate	N/F	73	Quinic acid	13.9	
30	Eriocitrin	N/F	74	Rhoifolin
(apigenin 7-O-neohesperidoside)	N/F	
31	Eriodictyol (3,4,5,7-tetrahydroxyflavanone)	N/F	75	Rosmarinic acid	14.426	
32	Esculin hydrate	1.643	76	Rutin hydrate M-OH2	N/F	
33	Ethylgallate	N/F	77	Sakuranetin (naringenin 7-O-methyl ether)	N/F	
34	Etoposide	N/F	78	Salicylic acid	N/F	
35	Ferulic acid	N/F	79	Schaftoside	837.932	
36	Fisetin hydrate	2.304	80	Sinapic acid	N/F	
37	Formononetin (neochanin)	N/F	81	Syringic acid	14.118	
38	3,5,7-Trihydroxy-2-phenyl-4H-chromen-4-one
(galangin)	N/F	82	Tiliroside	N/F	
39	3,4,5-Trihydroxybenzoic acid (gallic acid)	10.499	83	trans-Cinnamic acid	3,331.911	
40	5,7-Dihydroxy-2-(4-hydroxyphenyl)-4H-chromen-4-one (apigenin)	N/F	84	Vanillic acid	35.353	
41	Genkwanin (4′,5-dihydroxy-7-methoxyflavone, apigenin 7-O-methyl ether)	N/F	85	Vanillin	29.377	
42	Gentisic acid	N/F	86	Vicenin 2	N/F	
43	Glabridin	N/F	87	Vitexin (apigenin 8-C-glucoside)	277.122	
44	Hesperidin	0.009	88	α-Cyano-4-hydroxycinnamic acid	N/F	
Note:

N/F, not found.

In this study, among the analyzed extracts, trans-cinnamic acid was the predominant compound with a concentration of 3,331.911 mg/kg. In contrast, hesperidin and neohesperidin had the lowest concentrations at 0.009 mg/kg. Our findings corroborate previous research indicating the potential therapeutic effects of cinnamic acid derivatives against various cancers, including lung, colon, and breast cancers (Wang et al., 2019). Orientin demonstrated robust antioxidant properties attributed to its potent radical scavenging activity, electrophilic index, high electron affinity, electron-donating ability, electronegativity, and adiabatic ionization potential (Lam et al., 2016). Narirutin, along with other flavonoids, exhibited notable antihypertensive, cholesterol-lowering, and insulin-stimulating properties (Mitra et al., 2022).

Furthermore, vitexin and isovitexin demonstrated anti-inflammatory properties that could mitigate myocardial ischemia-reperfusion injury (Ganesan & Xu, 2017). Caffeic acid, known for its robust antioxidant and antimicrobial activities, was identified as a promising therapeutic agent for treating dermal diseases, preventing premature aging, and enhancing collagen production (Magnani et al., 2014). Vanillin is an important metabolite frequently used in cosmetics, foods, beverages, and pharmaceuticals, exhibiting various biological activities including antimutagenic, antiangiogenic, anti-colitis, and antianalgesic (Ouelbani et al., 2020). These findings highlight the potential of natural compounds, such as those identified in our study, as promising candidates for developing novel therapeutic interventions against various diseases.

Antioxidant activity

Biological systems have intrinsic antioxidant defense mechanisms to mitigate free radical-induced damage. Plant-derived foods, functioning through reduction, free radical scavenging, and singlet oxygen scavenging mechanisms, are remarkable sources of antioxidants (Lee, Koo & Min, 2004). Research on plant antioxidant capacity often includes evaluations of their phenolic and flavonoid contents (Renda et al., 2019; Ouelbani et al., 2020).

In this study, the DPPH antioxidant activity of Bellevalia pseudolongipes plant extract, obtained via maceration using methanol solvent, was measured at 0.29 ± 0.2 mg/g Trolox equivalent, whereas the ABTS antioxidant activity was 117.39 mg/L Trolox equivalent. These results reveal both similarities and differences compared to existing findings in the literature.

Balos (2021) reported IC50 values ranging from 102.11 to 911.57 mg/L in DPPH antioxidant activity analyses performed on different parts of the B. pseudolongipes plant using water, hexane, and ethanol solvents. The most effective results were observed at 322.81 mg/L in the bulb part, 152.73 mg/L in the leaf part, and 102.11 mg/L in the flower part, all with ethanol as the solvent. Conversely, in our study, the antioxidant activity of the extract obtained using methanol solvent was determined to be 0.29 ± 0.2 mg/g Trolox equivalent, indicating a significant effect of the solvent on antioxidant activity.

For ABTS activity analysis, Balos (2021) found IC50 values ranging from 114.16 to 618.34 mg/L, with the best results being 193.15 mg/L in the onion part, 140.56 mg/L in the leaf part, and 114.16 mg/L in the flower part using ethanol. The ABTS activity analysis of our methanol-extracted sample was measured at 117.39 mg/L Trolox equivalent. These findings align with the documented ranges and underscore the influence of methodological variations on the results.

Khorasani Esmaeili et al. (2015) emphasized that the extraction method, plant parts, plant age, and environmental conditions considerably influence antioxidant activity. Tekin (2022) reported inhibition values of 97.14% in onion extract, 44.87% in stem extract, 67.25% in leaf extract, and 85.50% in flower extract during ABTS analysis performed using ethanol solvent on different parts of the Bellevalia sasonii plant. In the same study, DPPH analysis showed 92.51% inhibition values in onion extract, 83.16% in stem extract, 92.25% in leaf extract, and 92.83% in flower extract, highlighting the role of plant parts in determining antioxidant activity.

Pedis (2019) demonstrated that the antioxidant activity in Bellevalia crassa varies with concentration, and in our study, heightened concentrations correlated with increased antioxidant activity. Yildirim, Mehmet & Paksoy (2013) and Sanei et al. (2021) found high antioxidant activity in the leaves of Bellevalia gracilis and Bellevalia dichroa, respectively. These findings emphasize the strong antioxidant properties of B. pseudolongipes, largely attributable to the high phenolic content.

The study, however, has limitations. While the antioxidant activity of B. pseudolongipes was evaluated, comprehensive antioxidant profiling requires additional tests. The clinical significance of these results remains unconfirmed, necessitating further in vivo and clinical studies to better understand the effects of the antioxidant properties of B. pseudolongipes on human health. Moreover, the antioxidant properties of B. pseudolongipes should be investigated under different extraction methods and conditions.

Analytical methods and standards can vary, leading to the detection of different compounds and results discrepancies. Variations in plant samples can affect the presence and concentration of compounds. This study mainly focused on the pharmacological effects and therapeutic potential of the identified compounds. Therefore, further studies are needed to investigate the applicability and efficacy of these compounds in food, cosmetics, and other sectors. Finally, detailed studies into the pharmacokinetic and pharmacodynamic properties of the identified compounds, including bioavailability, toxicology, and interaction profiles, are critical for the safe and effective use of herbal extracts.

Conclusions

To the best of my knowledge, this study is the first comprehensive analysis of the chemical composition of B. pseudolongipes using LC–HRMS. The phytochemical analysis enabled the identification of 38 distinct compounds, including trans-cinnamic acid, caffeic acid, vitexin, schaftoside, orientin, and narirutin. Collectively, the results highlight the diverse phytochemical composition and robust antioxidant properties of B. pseudolongipes, creating opportunities for potential applications in medicine, food, cosmetics, and various other industries. Nevertheless, further research is essential to thoroughly investigate the effect of B. pseudolongipes on human health, to gain more insight into its toxicological profile, and fully exploit its industrial potential.

Supplemental Information

Supplemental Information 1 Quantitation Report.

Supplemental Information 2 Analysis Report and results.

Supplemental Information 3 Calibration Report.

Supplemental Information 4 Confirmation Report.

Supplemental Information 5 Antioxidant activity results.

Additional Information and Declarations

Competing Interests

Author Contributions

Data Availability

The author declares that they have no competing interests.

İdris Yolbaş conceived and designed the experiments, performed the experiments, analyzed the data, prepared figures and/or tables, authored or reviewed drafts of the article, and approved the final draft.

The following information was supplied regarding data availability:

The raw data is available in the Supplemental File.

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
