# Peer review of "Phytochemical profiling and antioxidant activity assessment of Bellevalia pseudolongipes via liquid chromatography-high-resolution mass spectrometry"

_PeerJ, doi:10.7717/peerj.18046_

## Round 0.1 · original submission · Major Revisions

As you can see, both the reviewers recommended requested further modifications. Therefore, the manuscript cannot be accepted as is. However, I am willing to review my decision if you respond to the reviewers' requests. I ask for special attention to the issues raised by reviewer #1.

Reviewer 1 ·

Basic reporting

The “Phytochemical Analysis (Liquid Chromatography-High-Resolution Mass Spectrometry) and Determination of Antioxidant Activity of Bellevalia pseudolongipes (Asparagaceae)”, authored by İdris Yolbaş describes the chemical composition of B. pseudolongipes using liquid chromatography–high-resolution mass spectrometry (LC-HRMS) and the antioxidant effect in DPPH assay. The protocol used in chemical characterization agrees with literature procedures. However, the study seems incomplete to me. I suggest adding new experiments to better evaluate the antioxidant potential, as well as studying the toxicological effect of B. pseudolongipes. This work could be a relevant contribution for chemistry of natural products, but I don’t recommend the acceptance of this manuscript in Peer J. this way.

Experimental design

The research agrees with the scope of the journal. However new experiments must be included to complement the work and outline possible applications.

Validity of the findings

The results obtained are promising. However, they are not enough for publication. The study needs to be complemented.

·

Basic reporting

The manuscript, Phytochemical Analysis (Liquid Chromatography – High Resolution Mass Spectrometry) and Determination of Antioxidant Activity Bellevalia pseudolongipes (Asparagaceae) has been thoroughly read and the following points noted:
First and foremost, it will be more appropriate if the title is rephrased owing to the fact that the bracketed subject matter does not corelate with the preceding words.
The comparisons in line 201 should fall within the same class or species.
The literature review of line 153-163 is interfering with the result discussion. It should be aligned with the introductory part and the same should be done for line 203-211
Line 219 should be analytical instead of analysis
The comparison of line 189 with 197-198 is not coherent because the work did not spell out the plant part(s) that was investigated so should not be compared with the other that the plant parts were clearly indicated

Experimental design

Line 73 -77 talks about collection of plant material but does not clearly explain the part of plant that was collected.
The location within the region where the plant was collected should also be stated

Validity of the findings

Line 142 says chemical compositions while the title is saying “phytochemical analysis”. There should be clarity on your work and this makes it mandatory for the title to be rephrased. Your results clearly shows the chemical components owing to the fact that the two words (chemical compositions and phytochemicals) are not exactly the same scientifically.

Additional comments

Manuscript can be published after revisions.

---

## Round 0.2 · accepted · Accept

Please, include the suggestions of reviewer #2 in the proof.

Reviewer 1 ·

Basic reporting

The “Phytochemical Profiling and Antioxidant Activity Assessment of Bellevalia pseudolongipes via Liquid Chromatography-High-Resolution Mass Spectrometry” authored by İdris Yolbaş describes the chemical composition of B. pseudolongipes using liquid chromatography–high-resolution mass spectrometry (LC-HRMS) and the antioxidant effect in DPPH and ABTS assay. The protocol used in chemical characterization agrees with literature procedures. After evaluating the changes made to the manuscript by the author, which were requested in the first review, I recommend the acceptance of this manuscript in Peer J.

Experimental design

The research agrees with the scope of the journal.

Validity of the findings

The results obtained are promising and may have applications in different areas of knowledge.

Additional comments

Considering that the antioxidant activity of the extract was evaluated in the DPPH and ABTS assays, I suggest changing the title within the "Results and discussions" item. I suggest changing "DPPH radical scavenging activity" (Line 212) to something that considers both experiments. "Antioxidant activity", for example.

·

Basic reporting

The reviewed manuscript titled; Phytochemical Profiling and Antioxidant Activity Assessment of Bellevalia pseudolongipes via Liquid Chromatography-High-Resolution Mass Spectrometry by İdris Yolbaş has been read through once again. The writing and language is clear, understandable and concise.

The corrections formally pointed out has been adhered to and perfectly carried out as cross-checked. Literaure references and enough background provided as required within the scope of the Journal.
An important observation however, is not italicizing the "et al" word since it is not an English word and the plant name wherever it appears..

Experimental design

The research appears to fill the gap that it emphasized about having identified naturally occuring phytochemicals as antioxidants but the toxicological effects of same would have gone a long way to justifyng the effect on humans. This can be included in the conclusion as part of the recommendations for further studies..

Validity of the findings

The results as seen can be replicated and conclusions connected to the original question.

Additional comments

"|Therefore, the aim of the present study is to comprehensively analyze the chemical composition and radical scavenging and antioxidant activities of B. pseudolongipe"

Remove the "and" after composition.